# Selected Techniques for Physiotherapy in Dogs

**DOI:** 10.3390/ani12141760

**Published:** 2022-07-08

**Authors:** Marta Dybczyńska, Małgorzata Goleman, Aleksandra Garbiec, Mirosław Karpiński

**Affiliations:** Department of Ethology and Wildlife Management, University of Life Sciences, Akademicka 13, 20-950 Lublin, Poland; marta.dybczynska@up.lublin.pl (M.D.); aleksandra.garbiec@up.lublin.pl (A.G.); miroslaw.karpinski@up.lublin.pl (M.K.)

**Keywords:** canine, physiotherapy, animal nursing, animal physiotherapy

## Abstract

**Simple Summary:**

Animal physiotherapy is widely addressed in many scientific reports. Although the effectiveness of physiotherapy has been repeatedly proven in both human and animal models, a small percentage of animal keepers decide to implement physiotherapy as part of treatment programs for their pets. After horses, dogs are the second group of animals that are most frequently subjected to physiotherapy treatments. The present study compiles the current knowledge of canine physiotherapy methods. An attempt was made to systematize this knowledge through selection and description of the most important aspects of canine physiotherapy. Most of the physiotherapeutic methods have been transferred from human treatment protocols. A key issue in the achievement of therapy success is the proper selection of physiotherapeutic procedures and close cooperation between the veterinarian and the physiotherapist.

**Abstract:**

Physiotherapy is a new dynamically developing field of science in which the original idea was to improve the care for convalescent patients. Its positive effects observed in humans suggested the need for the adaptation and implementation of human physiotherapy techniques in animal care. Dogs are the second group of animals that undergo physiotherapy most frequently. These animals are diagnosed with a number of locomotor system problems, which may be congenital and are often related to the breed or acquired. The aim of the study was to collect and systematize knowledge of animal physiotherapy with emphasis on the selection and description of the most important aspects of canine physiotherapy. The review material consisted of 59 publications, with 230 selected for the review. Physiotherapeutic treatments are applied not only for rehabilitation of animals but also in healthy animals to upgrade their sports performance and improve their welfare. A majority of physiotherapeutic approaches have been transferred from human protocols. A key issue in the achievement of therapy success is the proper selection of physiotherapeutic procedures and close cooperation between the veterinarian and the physiotherapist.

## 1. Introduction

Physiotherapy is a dynamically developing field of medical science that was originally associated with orthopedics and neurology [1]. The gradual development of the methods for rehabilitation and support of the treatment of musculoskeletal diseases and neuropathies led to the emergence of a separate physiotherapist profession [2]. The task of a physiotherapist is to assess the patient’s motor dysfunctions and to plan and apply appropriate therapeutic techniques [3]. In principle, therapeutic activities focus on alleviation of pain and the improvement of the quality of patient’s life; additionally, they may also support the healing process [4]. Currently, physiotherapy is based on almost all available forms of non-pharmacological impact on the patient, i.e., physical therapy, kinesiotherapy, and manual therapy [5].

The large number of available scientific reports proving the effectiveness of physiotherapy has contributed to the development of analogous procedures, i.e., the adaptation and implementation of human physiotherapy techniques, in animal care [6]. Although its effectiveness and benefits have been repeatedly proven, animal physiotherapy is still underestimated but deserves more attention considering its promising future [7]. Animal physiotherapy is not only provided to companion animals, but is now successfully used in farm animals and, in exceptional cases, in wild, live animal species, e.g., those kept in zoos. Physiotherapy based on protocols tailored to individual needs can be applied in human and animal patients of all ages [1].

At present, horses are subjected to physiotherapy most frequently, which is closely related to the requirements established by their owners. Equestrian sports require considerable physical fitness of these animals as well as freedom from pain and any discomfort during movement [8]. Dogs are the second most frequent group of physiotherapy patients [5]. These animals are diagnosed with many locomotor problems, which may be congenital (e.g., susceptibility to a given disease related to the breed) or acquired [9]. The recent development of canine sports has contributed to the increased interest in canine physiotherapy [10]. In sports dogs, physical treatments are aimed at improvement or maintenance of their physical fitness [10]. However, the assessment of the effectiveness of therapeutic treatments in dogs is difficult due to their status as companion animals, which in most cases allows the use of only non-invasive research methods. Hence, the available scientific reports are often limited to presentation of “case studies”. Such results are difficult to relate to the entire population since the species *Canis familiaris* comprises approximately 400 breeds with varied morphology and different predispositions to the pathology of the locomotor system. The present review presents the available literature on animal physiotherapy with special focus on canine physiotherapy in order to collect and systematize the knowledge in this field.

## 2. Materials and Methods

The review material comprised 59 scientific articles selected from a collection of 230 papers on animal physiotherapy. The articles were found in the following databases: Google Scholar, Web of Science, Scopus, PubMed, Wiley Library, and ResearchGate. The databases were searched in the order listed above. The research included only articles available in English.

The selection of scientific reports on the relevant topics was based on the search for the following keywords: massage, dog, canine, physiotherapy, animal physiotherapy, kinesiotherapy, hydrotherapy, taping, kinesiotaping, relaxation, laser therapy, impact, effect, introduction, magnetotherapy, electrotherapy, prolotherapy, behavior, guided exercise, evaluation, validation, therapeutic, ultrasound, acupuncture, animal, pain management, treatment, rehabilitation, care, nursing, and ultrasonic.

The phrases mentioned above were used both as phrases and as separate plural and singular words. All authors participated in full-text screening; each of them independently selected a collection of valuable publications. Only articles that were approved in each of the selected collections are included in this paper.

As many as 171 articles found were rejected based on the following criteria:papers older than the year 2000 (the authors intended to present the latest state of knowledge);abstracts indicating different research problems addressed (e.g., physiotherapy was a secondary subject in the paper);investigations of a small number of individuals (the authors assessed the credibility of the study by the number of participants, which does not include case study papers).

The quality of the studies was assessed by the following factors:The number of participants (a higher number usually means a better quality of study);More than one study method/physiological indicator (this allows comparison of the results);The citation rate in the case of reviews.

The selection of articles proceeded as shown in Figure 1.

The remaining 59 papers were divided into the following groups:Review articles 30.51%;Original research articles 59.32%;Articles on related issues (e.g., animals other than dogs) with valuable content to be presented in the review 10.17%.After selecting the articles for analysis, each text was screened for data on:Indications for physiotherapy;Types of physiotherapy described in the article;Proven effectiveness of the described therapy;Research methods;Comparisons with physiotherapy used in other species.

The results obtained by screening the available data are presented below.

## 3. Results

Most (77%) of the analyzed original studies are high-quality research works. Review articles that are often cited in the literature on animal physiotherapy were selected for this work as crucial contributions. The precise eligibility criteria for the review as well as the independent work of all authors allowed decreasing the risk of bias.

The authors considered three major indications for physiotherapy mentioned in the articles. The post-traumatic group comprises indications related to the process of healing injuries. Age-related indications were degenerative changes caused by the aging of patients. Physiotherapy for healthy animals was regarded as a way to prevent injuries and improve the fitness of the animal.

Indications for physiotherapy presented in the available papers:−Post-traumatic—in 54.24% (32) of publications;−Age-related degeneration—in 35.60% (21) of publications;−Physiotherapy in healthy animals—in 23.73% (14) of publications.

Most papers present more than one indication for physiotherapy. The overwhelming majority of the publications analyze post-injury cases of physiotherapy care. Many articles describe age-related indications for physiotherapy, which is probably caused by the frequently diagnosed degenerative disease osteoarthritis. Only a small percentage of the publications mention indications for the physiotherapy of healthy dogs, and most of them are review articles. It is important to emphasize the number of articles discussing particular physiotherapy procedures (Figure 2).

The hydrotherapy procedure is described by 22% of the analyzed articles, which can be an indicator of the frequency of implementation of this type of physiotherapy in practice. This method has gained considerable popularity over the last few years. Hydrotherapy includes both treadmill walking and swimming. To ensure safety of this type of therapy, the hydrotherapist should obtain complete information about the patient’s condition from the veterinarian [11]. For instance, epilepsy, diarrhea, open wounds, or dermatological diseases may be contraindications to hydrotherapy. Patients with aquaphobia and claustrophobia should not undergo hydrotherapy either [12]. This type of therapy is appreciated in the treatment of such incurable chronic diseases as osteoarthritis [13,14]. Exercise in water provides increased buoyancy, hydrostatic pressure, and increased resistance, which change the work of the patient’s locomotor system and reduce the load on the animal’s joints [15,16,17]. Hydrotherapy facilitates exercise with a reduced body weight load, which increases the range of motion in the joints and stimulates the growth of muscle mass and nerve endings [15,18,19]. It also contributes to improvement of lymphatic drainage in patients with edema. An important issue in water treadmill–based therapy is the depth of the water. A higher water level puts more resistance on the extremities and forces increased effort, which should be considered at the stage of planning the hydrotherapy scheme [20].

Another frequently used therapy is heat and cold treatment. Different sources of temperature change are used, but water in various states of aggregation is often used in this procedure. Local cooling can be helpful in the treatment of swelling, alleviation of pain, or local vasoconstriction. The highest effectiveness of this treatment is achieved within the first 72 h post-trauma [21]. Selected injured body areas are locally cooled in this way most often, but short-term cooling of the entire body in a cryo chamber is applied in physiotherapy as well [22]. In humans, the whole body is subjected to cryotherapy, whereas the head of animal patients cannot be exposed to this procedure. Cryotherapy has been widely used in human and animal oncology [23]. In turn, warm compresses are applied topically to treat stiffness, rheumatic ailments, and muscle contracture. 

Magnetic field therapy is popular in both human and animal physiotherapy. It can be based on constant magnetic field or alternating electromagnetic pulses. The constant magnetic field has an impact on the processes of healing and regeneration of the osteochondral system by prevention or delay of necrosis and apoptosis of mesenchymal cells. As evidenced by Marędziak et al. (2014) [24] and Sayfzadeh et al. (2007) [25], a slowly varying magnetic field has a positive effect on the rate of bone fracture repair.

Rogachefsky et al. (2004) [26] reported that exposure to a constant magnetic field can inhibit the progression of osteoarthritis. Alternating magnetic field therapy, mainly applied locally to increase blood flow to the treated organ, has a number of applications, e.g., in the treatment of prostate hyperplasia caused by blood circulation disorders [27].

Laser therapy is an interesting and relatively new method of animal physiotherapy used in the treatment of wounds and inflammation. As shown by Ruffoni (2017) [28], laser therapy used to irradiate acupuncture points can bring positive effects in the treatment of behavioral problems. Laser biostimulation with varied intensity is aimed at the acceleration of healing and regeneration processes in soft tissues. Various investigations have confirmed that the laser wave increases the number of active fibroblasts in the treated area and stimulates the formation of new collagen fibers [29]. Wardlaw et al. (2019) [30] investigated the effect of laser therapy on postoperative wound healing in dachshunds. The laser stimulation not only accelerated the healing process but also had a positive effect on the cosmetic appearance of the scar. Olivierii et al. (2015) [31] conducted a pilot study to assess the effect of low-level laser therapy on hair regrowth in cases of non-inflammatory alopecia. The results of the study turned out to be promising, as all dogs exhibited improved hair regrowth. This indicates that laser therapy can have a very wide range of applications, depending on the therapeutic protocol.

Neurostimulation consists in transcutaneous or direct stimulation of neurons with low-intensity currents. Such stimulation contributes to elimination of pain through disruption of pain impulses in the nervous system. Direct stimulation of muscles with electric impulses is used in, e.g., rehabilitation after orthopedic surgeries to restore the natural muscle contraction mechanism and in the electro-acupuncture treatment of spastic patients. The presence of a pacemaker, epilepsy, advanced vascular disease, and reduced pain sensation may be contraindications to this type of therapy [32].

Shockwave therapy, which is often applied in human physiotherapy, is much less frequently used in the treatment of animals. The procedure is based on the use of a precisely targeted high-energy sound waves with specific properties. The deep tissue penetration helps to reduce pain, dissolve calcium deposits in tissues, and accelerate tissue regeneration. This technique has been transferred from human physiotherapy as a treatment for tendon injuries in horses, which is a common problem in sport horses as their mucoskeletal system is subjected to enormous strain. The access to tendons is difficult as they are anatomically located close to bones or are covered by bones. This type of physiotherapy has been successfully used in horses to relieve pain in the thoracic–lumbar spine and to reduce the concentration of inflammatory biomarkers [33,34]. This suggests a possibility to implement shockwave therapy in dogs, although there are fewer scientific reports on its effectiveness in this animal species than in horses. In their study on eight dogs with acute femoral fracture, Wang et al. (2001) [35] reported that this method accelerated bone repair substantially. In the case of stifle joint osteoarthritis in dogs, shockwave therapy was found to exert a positive effect on the range of motion in the treated animals [36]. Ultrasound therapy is more effective in dogs than in horses, which is related to the different ranges of sounds heard by the two species. The shockwave sound may frighten dogs, unlike ultrasound, which is not audible. It has been evidenced that low-frequency ultrasound therapy significantly accelerates repair processes in irradiated bone in the treatment of mandibular osteoradionecrosis [37]. Miyamoto et al. (2003) [38] tested the possibility of pharmacological substitution of coronary vasodilatation with low-frequency ultrasound in a canine model. Blood vessels were found to dilate within a second of the application of the therapy at a level similar to that of nitroglycerin administered to patients.

Various types of massage are the most commonly reported procedure of physiotherapy which is consistent with observed practice. The massage consists in manipulation of tissues, muscles, and fascia performed both manually and with the help of mechanical vibration massagers. In human studies, massage has been shown not only to have an impact on blood circulation and release of post-exercise metabolites but also to bring relaxing effects [39]. The healing effect of massage is helpful in managing swelling as it helps to drain the excess of accumulated lymph [6]. The relaxing effects of massage include a slower breathing rate, a lower heart rate, reduced muscle tension, and such biochemical effects as reduced cortisol concentrations and elevated serotonin levels in the blood of patients [40,41,42]. The massage techniques can be divided into stroking, effleurage, compression, friction, fascial manipulations, and strictly relaxing massage [41]. Massage can quickly improve the comfort and range of motion in dogs, which is reflected in behavioral changes [42]. Rilley et al. (2021) [43] observed behavioral pain indicators, gait, posture, and activity in a group of 527 dogs. Based on the assessment carried out by both the dogs’ owners and physiotherapists, the researchers found that therapeutic massage significantly reduced pain perception by the animals. In many cases, including, e.g., hip dysplasia, it is beneficial to combine massage with other therapeutic methods, e.g., cooling [44]. Massage in the form of deliberate manipulation of joints can be extremely helpful in managing canine osteoarthritis. It contributes to increased production of synovial fluid and alleviates pain [45]. A difficult issue is the management of geriatric and palliative animal patients. Pressure ulcers are the most common progressive problem caused by immobility. Massage can replace the natural pressure of muscles on soft tissues and vessels occurring during movement [14]. As reported by Fuentes Beneytez M. (2021) [46], the regular Chinese traditional tui-na massage increases the effectiveness of pharmacological treatment in dogs with osteoarthritis. In addition to its healing properties, massage can be helpful in managing a number of behavioral problems related to stress and tension. There are many scientific reports on the application of this relaxation technique in humans. The assessment of changes in the mental comfort in dogs is much more difficult. Since dogs are companion animals, invasive methods for investigations of these animals are restricted. Therefore, only indirect indicators of stress, i.e., heart rate, number of breaths, and saliva cortisol level, can be evaluated in dogs [47].

Guided exercises are one of the most frequently used methods in dog physiotherapy due to not only their effectiveness but also the possibility to be implemented by an appropriately trained owner in home conditions [48]. The exercises can be divided into movement forced by the handler and that performed independently by the dog on command. The exercises performed at home as a complement to therapy should be simple and short, with emphasis on their quality and correctness [45]. A dog with musculoskeletal diseases most often tries to save the sore limb, which may lead to the atrophy of temporarily unused muscles and strain the healthy limb, inducing secondary trauma. Therapeutic exercises prevent loss of muscle mass and accelerate post-trauma recovery [49]. Regular warm-up or cool-down exercises performed after training help to prevent injuries in dogs practicing canine sports. Core stability exercises stimulate the development of muscles responsible for the body posture and protect the joints and backbone from overload. In some sports, such as obedience, the dog is expected to maintain an unnatural body posture for a longer time, which may overload some muscles; hence, physiotherapy may serve as a form of prophylaxis in clinically healthy dogs [50]. 

Kinesiotaping is often used to support manual physiotherapy or kinesiotherapy. Tapes applied properly onto the patient’s body relieve strained tissues. There are several methods of application of the tapes, with one of them supporting the absorption of edema and hematomas. Kinesiotaping is widely and successfully used in human and equine physiotherapy [51]. Tapes with properly selected adhesiveness can be used in many animal species. This method is mostly appreciated by the owners of sports dogs as the tapes do not exclude the dog’s movements or bathing, and the sensitive area of tissue is constantly protected at the same time.

Acupuncture involves the insertion of needles into precisely defined points on the skin called meridians. Although it originates from alternative medicine, there are increasing numbers of scientific reports on its therapeutic effectiveness. Acupuncture assisted by low-intensity electrical impulses is referred to as electro-acupuncture [52]. In a study of 181 dogs with neurological and musculoskeletal disorders, Silva et al. (2017) [53] demonstrated that acupuncture reduced pain and improved the quality of life of patients with these types of diseases.

All the articles under consideration report positive effects of the discussed types of physical therapy, although their quality varies (Table 1).

## 4. Discussion

Human and animal physiotherapy is often associated exclusively with the treatment of orthopedic or neurological conditions. In fact, properly implemented physiotherapy can help patients suffering from a variety of ailments. In practice, physiotherapeutic procedures are often intended for patients who are unable to move independently or move in a pathological manner [54,55]. Such a condition in animals may result from previous traumas or chronic neurological diseases. It may also be associated with the period of convalescence after specialist procedures or may be a consequence of diseases exerting a secondary effect on the locomotor system, e.g., obesity [55]. The effectiveness of physiotherapy largely depends on adequate cooperation between the physiotherapist and the veterinary doctor. Such effective teamwork is possible when the doctor can appreciate the benefits of physiotherapy applied as part of patient treatment, and the competence of the physiotherapist is complementary to that of the clinician [56]. Another aspect that should be considered in the assessment of the effectiveness of physiotherapy is its appropriate adaptation to the needs of a specific case. One of the growing problems nowadays is the care for geriatric animals. Not only the species but also the breed, size, physical activity, and previous or ongoing diseases are the determinants of the aging process rate in the case of dogs. Physiotherapy is often implemented to improve the quality of life of geriatric patients or to prevent premature euthanasia [57].

Physiotherapy clinics offer a wide range of specialist treatments, but the key issue is to choose the appropriate method suitable for the patient. In general, physiotherapy can be divided into:Physical therapy—the use of physical factors, e.g., water, electricity, magnetic field, light pulses, etc.;Manual therapy—hand therapy; usually various types of massage;Kinesiotherapy—movement exercises performed by the patient under the supervision of a physiotherapist.

It is impossible to identify the most effective physiotherapy method. The number of scientific reports on a given type of physiotherapy and the frequency of its implementation do not correspond with each other. The success of a particular type of therapy depends on many variable factors. In addition to general contraindications, many individual breed-related traits and dog behaviors should be considered.

The most innovative methods implemented in animal physiotherapy increasingly often resemble those used in veterinary medicine. Tae-Hwa K et al. (2006) [58] managed to eliminate symptoms of right hip joint osteoarthritis and atrophy of the surrounding muscle in a 6-year-old Pointer dog within about a month. The researchers injected bee venom into five acupoints on the affected limb. Acupuncture has been regarded as a physiotherapeutic technique for years on the one hand, but, on the other hand, the injection of foreign substances into the patient’s body can be qualified as a medical procedure.

There are many scientific reports focused on elucidation of the biological mechanism of treatment of epileptic seizures with properly applied acupuncture. Researchers agree that this method brings excellent results and most probably consists in the modulation of the secretion of neurotransmitters responsible for seizures [59]. Still, acupuncture is considered alternative medicine.

This article also highlights gaps in the literature. There are techniques of physiotherapy, successfully used in animal physiotherapy, which the authors have observed from experience, and yet there are no scientific reports on their effectiveness in given species that could be referred to in this paper.

## 5. Conclusions

With the progress in animal physiotherapy, increasing numbers of the techniques used in human therapy are adapted to be suitable for animal treatment. Many of them have to be modified and adapted to the needs of a specific animal species. Canine physiotherapists usually choose manual therapies, hydrotherapy, ultrasound, or guided exercise. Innovative treatment methods often combine medical intervention with strictly physiotherapeutic procedures; therefore, cooperation between the veterinary doctor and the physiotherapist is extremely important. The rapidly expanding spectrum of physiotherapy effects achieved in dogs gives hope for higher effectiveness of treatment and rehabilitation in the future. Nevertheless, the frequency of the implementation of specific therapies is not reflected in the number of scientific reports of their effectiveness. This is related to the difficulty in the assessment of the effects of different therapy types in animals. Examinations of dogs with invasive methods are subject to local ethics committee restrictions. The vast majority of studies describe the effects of physiotherapy in diseased or injured animals, whereas a small percentage focuses on the implementation of physiotherapy in healthy animals to improve their fitness and performance. This method for improvement of the condition of animals is becoming increasingly popular, especially in the case of sports dogs.

## Figures and Tables

**Figure 1 animals-12-01760-f001:**
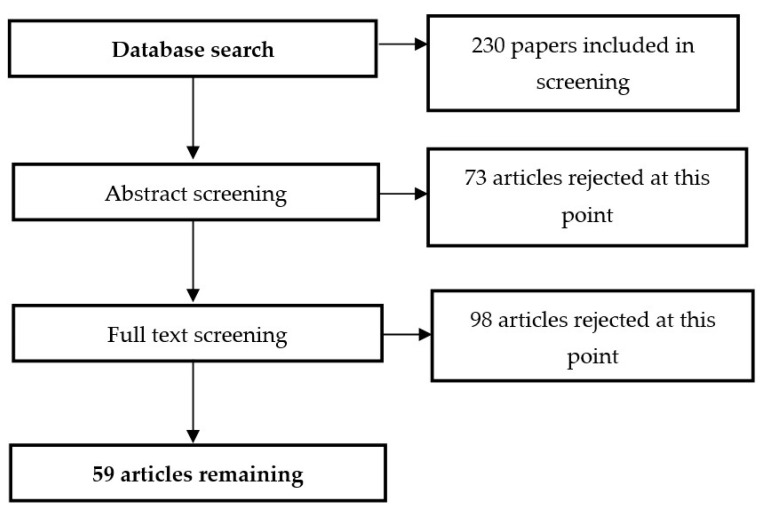
Selection of articles.

**Figure 2 animals-12-01760-f002:**
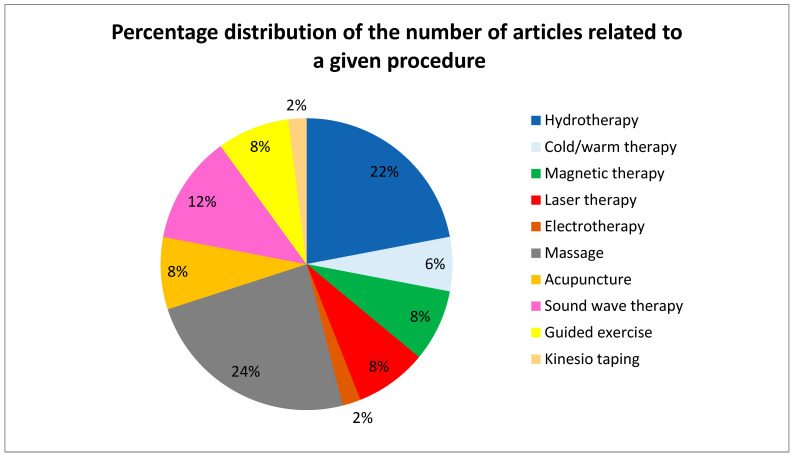
Percentage distribution of the number of articles related to a given procedure.

**Table 1 animals-12-01760-t001:** Juxtaposition of data from original research articles.

Number of Citation	Species	Number of Participants	Health State	Type of Physiotherapy	Methods	Effectiveness
11	Dogs	83	Postoperative with undergoing thoracolumbar hemilaminectomy	Hydrotherapy	Neurological and physical examination; also follow up visits	Successful
13	Dogs	55	Osteoarthritis	Hydrotherapy	Evaluation of the level of biomarkers for osteoarthritis	Most parameters showed improvement
20	Dogs	8	Healthy	Hydrotherapy	Kinematic analysis	Observed changes related to the water depth
21	Dogs	12	Removal of undesirable bronchial mucosa tissue	Cryotherapy	Bronchoscopy and histological evaluation	Successful
22	Dogs	3	Persistent papilloma	Cryotherapy	Histological evaluation	Successful
24	Canine and equine cells	In vitro	Healthy cells	Magnetic field	Morphological changes in mesenchymal stem cells derived from adipose tissue	Positive morphological changes
25	Dogs	10	Bone healing	Magnetic field	Evaluation of radiographic and histopathological changes	Positive radiographic and morphological changes
26	Dogs	18	Osteoarthritis	Magnetic field	Assessment of anatomical and morphological changes	Successful inhibition of development of osteoarthritis
27	Dogs	20	Prostatic hyperplasia	Magnetic field	Doppler assessment by ultrasonography, libido, semen quality, testosterone levels, and seminal plasma evaluated	Successful
30	Dogs	9	Incision healing	Laser therapy	Scar measurement and observation of the healing process	Successful
31	Dogs	7	Noninflammatory alopecia	Laser therapy	Coat regrowth observation	Successful
33	Horses	12	Thoracolumbar pain	Shockwave therapy	Ultrasound examination of muscle and palpation score of the pain	Successful
35	Dogs	8	Acute Fractures of the Tibia	Shockwave therapy	Radiographic and histological examination	Successful
36	Dogs	10	Osteoarthritis	Shockwave therapy	Kinematic examination	Successful
38	Dogs	12	Healthy	Ultrasound therapy	Coronary vasodilatations assessed by ultrasound and angiography	Successful
42	Dogs	47	Musculoskeletal pain	Manual therapies and acupuncture	Questionnaire	Successful
43	Dogs	527	Pain	Massage therapy	Changes in the number and severity of issues for five pain indicators and quality of life score	Successful
46	Dogs	30	Osteoarthritis	Massage therapy	Isokinetic muscle strength measurements and pain questionnaire	Successful
49	Dogs	8	Healthy	Exercise therapy	Kinematics assessment (range of motion, flexion, and extension)	Positive changes compared to the control group
51	Horses	Case study	Healthy	Various types	assessment of training effects	Successful
53	Dogs	181	Neurological and musculoskeletal diseases	Acupuncture	Medical assessment and pain score	Successful
58	Dogs	Case study	Osteoarthritis	Acupuncture with bee venom	Radiographic evaluation	Successful

## Data Availability

Not applicable.

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
