# Peer review of "Selected Techniques for Physiotherapy in Dogs"

_animals, 2022, doi:10.3390/ani12141760_

Round 1

Reviewer 1 Report

The manuscript „Comparative analysis of the potential to implement some physiotherapy types in dogs” is a review work. It concerns a new field in veterinary and animal science - animal physiotherapy. The authors focused on the species Canis familiaris. The aim of the study was to collect and systematize knowledge of zoophysiotherapy with emphasis on the selection and description of the most important aspects of canine physiotherapy. The articles were selected using the following databases: Google Scholar, Web of Science, Scopus, PubMed, Willey Liberty,and  Researchegate, using appropriate keywords ((massage, dog, canine,  physiotherapy, zoophysiotherapy, kinesiotherapy, hydrotherapy, taping, kinesiotaping, relaxation, laser therapy, impact, effect, introduction, magnetotherapy, electrotherapy, prolotherapy, behavior, Guided exercise, evaluation, validation, therapeutic, ultrasound, acupuncture, animal, pain management, treatment, rehabilitation, care, nursing, ultrasonic).

The authors searched for and reviewed 230 publications on physiotherapy and various methods of its application, from which they selected 59 articles for analysis and review. The number of analyzed publications seems to be fully sufficient to explore the topic of canine physiotherapy, considering it is a relatively new subject. The development of cynology and the use of dogs in sports means that owners of healthy dogs often want to use physiotherapeutic treatments to improve the condition and sports form of their dogs. Many forms of physiotherapy used in other animal species, e.g. horses, and human physiotherapy can be transferred to dogs. However, this requires adapting the method to the species of dog. I believe that the authors' list of canine physiotherapy methods is a very important and practical report and it is a topical subject.

The quantitative consideration of used articles, divided into types of physiotherapy, in the Material and methods chapter, and an extensive discussion are undoubted advantages of the manuscript.

Line 96-97 - In Figure 1, I do not see the percentage of articles on ultrasonic being included

The manuscript may be published in Animals journal after minor revision

Author Response

Answer for the Reviewer  1

Thank you very much for your thorough review and we appreciate your valuable comments.

In line with the comments of the reviewer, changes were made to the text and the title was modified.  Referring to the remark in Figure 1, ultrasounds were included in the "sound wave therapy" item because they constituted a very small percentage of the analyzed articles and were not visible on the chart.

Reviewer 2 Report

Thank you for the invitation to peer-review your work. I am a systematic review author with the Cochrane Collaboration, and this experience informs my comments. 

It is unclear to me whether you intended your manuscript to be a narrative review (referenced commentary) or a systematic review. I have presented my suggested refinements in two parts - please make a clear decision as to the type of review you are writing, and adopt the relevant guidance.

Narrative review

If your manuscript is intended as a narrative review, then the traditional scientific paper format including methods, results, and discussion is unnecessary. Narrative reviews are well referenced arguments, essays if you will, not research projects, thus no reader will need to use your methods section explaining how you identified resource materials because they will not be trying to replicate your study. If your intent was a narrative review, please re-write your manuscript in essay style. 

OR

Systematic review

If intended as a systematic review, then the method section lacks sufficient detail to allow replication of the research process, and the results and discussion sections require refinement.

Please add full details to the methods to explain which databases were searched, in which order. Clarify all inclusion and exclusion criteria for selected studies, and justify these decisions (eg: why limit studies to those published prior to 2000?). Provide details of methods used to grade the quality of included studies, and assess them for risk of bias. Outline your methods for data extraction from included studies, as well as any planned statistical analysis or meta-analysis. Include a PRISMA flow diagram to explain the stages of study selection. 

In the results section, add results of study quality grading and risk of bias assessment. Also, please add a table or summary of the effect sizes / outcomes identified with the use of the physiotherapy methods reported in the included studies. Without results regarding effect sizes or outcomes, your manuscript appears to be a popularity poll of canine physiotherapy techniques, perhaps erroneously equating treatments that are popular or common with those that are effective. 

Please refine the discussion section substantially. In the current format, the discussion section appears to be a narrative reporting of the included studies, which is not actually discussion as much as results. In a systematic review emphasis is given to the most robust conclusions, which are those that can be drawn from the aggregation of data across systematically sourced studies. Assertions drawn from single sources are afforded less weight. Please re-write this section to clarify how many studies inform each assertion, and provide details as to how many participants were included in each study, the procedures undertaken, and the outcomes obtained. Then add an overall statement or table to summarise the key findings from your review. 

Once this expanded results section is complete, you may wish to add a short discussion to place the results in the broader context of the field of study. 

You may also wish to add an overall statement or table to summarise the key findings from your review. 

There is a third type of review called a scoping review, which uses systematic methods, but includes studies across a wide range of designs. It is possible that you intended this manuscript to be a scoping review. Scoping reviews are usually undertaken to inform planned research. If this was your intention, please clarify the scope of your future research question to explain why the scoping review is required. Please refer to the above comments regarding systematic reviews, except for the notes on statistical analysis or meta-analysis, which cannot be undertaken when studies are of disparate designs. 

Author Response

Answer for the Reviewer  2

Thank you very much for your thorough review and we appreciate your valuable comments.

In accordance with the reviewer's comments, the manuscript was formulated as a systematic review. The abstract has been changed, the methodology and discussion have been improved, and the title has also been changed. All comments have been included in the text.

Some of the sentences in the discussion have not been changed because the sentences are not a quotation, but the result of the authors' experience. Such wording appears in many publications and is not quoted.

Reviewer 3 Report

The manuscript entitled “Comparative analysis of the potential to implement some physiotherapy types in dogs” is generally well-written, but I had a number of concerns regarding the data obtained and discussion.

Specific comments are followed below:

- Abstract

Lines 19-25 - The introduction in the abstract is too extensive. On the other hand, the results are too short.

Suggestion – Please, make the introduction short and expand on the results and conclusion.

1. Introduction

Line 37 - Suggested to exclude:  new - Physiotherapy is a dynamically developing field……..

All phrases must have an author (s).

Line 39 – “convalescent patients...” needs author.

Line 52 – “sub-discipline of veterinary medicine…” needs author.

Line 53 – Is zoophysiotherapy a term used with frequency in papers used to make this review? Usually, only physiotherapy and rehabilitation are mentioned... needs clarification.

Line 55 – Please include a period at the end of the sentence.

………….. those kept in zoos. Physio……….

Line 59 – “established by their owners…”; and Line 61 “Dogs are the second most frequent group of physiotherapy patients”. needs author. This information is very important to your paper. 

Line 64 – “interest in canine physiotherapy…”; Line 65 “their physical fitness…” needs author.

2. Materials and Methods - Some factors of inclusion and exclusion need clarification.

Line 76 – “zoophysiotherapy…” or Line 72 “…canine physiotherapy…” I think you need to standardize these terms in your paper! needs clarification.

Line 86 – Describe more about the 171 discarded works, plus factors that were considered for their exclusion.

Line 92 – “e.g. other animals…” other animals with valuable content? It's not just dogs? What other animals? need clarification.

Line 90 and others – Example: 30,51%  - correct in English – 30.51%

Please correct all in the text. The same problem in the results.

3. Results - Improve your results, you can show to us so much data for physio care in dogs. Please consider the table idea to better demonstrate your data.

Figure 1 must be linked to the text.

The results must be presented and not discussed. Example: In reality, not every physiotherapy center is equipped with a water treadmill, so this is not the most commonly used therapy. This part is a discussion.

Line 95 – “Indications for physiotherapy…” The general sum of publications used is equal to 67 (32 + 21 + 14), which happened with the 59 papers included. NOTE: If any paper has more than one indication, you need to explain.

Lines 96-99 – ABOUT INDICATIONS: What diseases are reported as post-traumatic? Which degenerative? It needs to describe and clarify, as the indications must be different.

Lines 100-104 - The analysis of the results is not clear; it does not show evidence of the indications of physiotherapy for diseases in dogs.

Line 102 – “degenerative disease…” Where? Which joints? Specific diseases? Hip dysplasia? Needs clarification; Line 103 – “indications for physiotherapy of healthy dogs…” They are agility competition dogs?... as well. I think you can explore more of the data obtained.

Lines 108-113 - If you could demonstrate the main types used for manual therapy? (massage), and how many used some equipment for the procedure of physiotherapy in practice your paper will be better.

NOTE: The figure for percentage distribution of the number of articles related to a given procedure is good, but doesn’t show to us the "Comparative analysis of the potential to implement some physiotherapy types in dogs"

NOTE: Maybe demonstrate the results in a table? Diseases / physio... treatments / Age / body weight/ breed of dog and other relevant information of papers included.

4. Discussion

Several phrases do not have an author. This is inadequate. Thus, each phrase must have an author or multiple authors.

You must link your results to the literature. The discussion must present the meaning of the results. 

Author Response

Answer for the Reviewer  3

Thank you very much for your thorough review and we appreciate your valuable comments.

*Lines 19-25 – the abstract has been modified

*Line 37 - Suggested to exclude:  new - Physiotherapy is a dynamically developing field……..

- We believe that this sentence is crucial, but it has been slightly modified in line with the reviewer's comments.

*Line 39 – “convalescent patients...” needs author

- This is the author's conclusion with an emphasis on "probably".

*Line 52 - “sub-discipline of veterinary medicine…” needs an author.

- Line changed, but this is not a citation. The authors are speaking from their experience (line 62 new revised manuscript)

*Line 53 – Is zoophysiotherapy a term used with frequency in papers used to make this review? Usually, only physiotherapy and rehabilitation are mentioned... needs clarification.

- All “ zoophysiotherapy “ changed in the text to Animals physiotherapy, following the vocabulary found in the articles

*Line 55 – done

*Line 59 – “established by their owners…”; 

-  Vocabulary changed but this is not a citation. The authors are speaking from their experience

*Line 61 -“Dogs are the second most frequent group of physiotherapy patients”. – a citation added

*Line 64 - a citation added

*Line 65 - a citation added

* Line 72 and 76  - all “ zoophysiotherapy “ changed in the text to Animals physiotherapy, following the vocabulary found in the articles

*Line 86 – Describe more about the 171 discarded works, plus factors that were considered for their exclusion. Line 92 – “e.g. other animals…” other animals with valuable content? It's not just dogs? What other animals? need clarification.

- The methodology has been revised as suggested by the reviewer

*Line 90 and others – Example: 30,51%  - correct in English – 30.51%

- In numbers, commas have been changed to dots

*Figure 1 must be linked to the text. – done (figure 2 in revised manuscript)

*The results must be presented and not discussed. Example: In reality, not every physiotherapy center is equipped with a water treadmill, so this is not the most commonly used therapy. This part is a discussion.

– Has been deleted

*Line 95 – “Indications for physiotherapy…” The general sum of publications used is equal to 67 (32 + 21 + 14), which happened with the 59 papers included. NOTE: If any paper has more than one indication, you need to explain.

– Has been described and clarified

*Lines 96-99 – ABOUT INDICATIONS: What diseases are reported as post-traumatic? Which degenerative? It needs to describe and clarify, as the indications must be different.

– Has been described and clarified

*Lines 100-104 - The analysis of the results is not clear; it does not show evidence of the indications of physiotherapy for diseases in dogs.  

– Has been changed

*Line 102 – “degenerative disease…” Where? Which joints? Specific diseases? Hip dysplasia? Needs clarification; Line 103 – “indications for physiotherapy of healthy dogs…” They are agility competition dogs?... as well. I think you can explore more of the data obtained.

- Has been described and clarified

*Lines 108-113 - If you could demonstrate the main types used for manual therapy? (massage), and how many used some equipment for the procedure of physiotherapy in practice your paper will be better.

- The authors wanted to maintain a balance between the described techniques of physiotherapy and the amount of text devoted to a given technique. A separate article could be written for each of the methods of physiotherapy.

*NOTE: The figure for percentage distribution of the number of articles related to a given procedure is good, but doesn’t show to us the "Comparative analysis of the potential to implement some physiotherapy types in dogs"

– The title has been changed

*NOTE: Maybe demonstrate the results in a table? Diseases / physio... treatments / Age / body weight/ breed of dog and other relevant information of papers included.

Such a juxtaposition would result in a very large chapter of results, which is not what this manuscript was about. The intention of the authors was to highlight the discussion and describe the physiotherapeutic methods, not to focus on statistics. Moreover, any statistical comparison is inadequate due to the use of different research methods within one physiotherapy technique and different sizes of research groups.

*Several phrases do not have an author. This is inadequate. Thus, each phrase must have an author or multiple authors.

- Some of the sentences in the discussion have not been changed because the sentences are not quotations, but the result of the authors' experience. Such wording appears in many publications and is not quoted.

*You must link your results to the literature. The discussion must present the meaning of the results. 

- In the result, we present a quantitative list of articles, not their content, so citing is not possible because it applies to all articles.

In accordance with the comments of the reviewer, all content which belonged to the discussion was removed from the results.

Reviewer 4 Report

Line 25. (and forward line 63). It is not clear how the term “congenital” in used. Are the authors referring to any specific disease? Need for explanations.

Line 32.  In this statement it is not clear the figure of the “physiotherapist” compared to the “veterinarian".  Maybe the author meant the “clinician” instead of the veterinarian? 

Close cooperation is surely needed. Better to explain that the cooperation must include all the different professional figures that play a fundamental role: the orthopedic clinician, the radiologist and the physiotherapist. Moreover the fourth figure involved is the owner that have to follow the indication of the med vet physiotherapist.

Line 51. This is not true. The proof is The college of sports medicine and rehabilitation that is rapidly growing up and rising. It is a reality. Maybe is better to say that physiotherapy deserves more attention in consideration of its promising future. 

Line 74. Author should better explain which are the inclusion criteria of selection on the papers.  The target of this paper is to present literature “in order to collect/systematize the knowledge in this field”. So it’s better to include  a more detailed declaration of inclusion criteria (I.e. indication , species, subject, procedures and method selected, pathologies included in the research…. And so on)

Line 101. This sounds like a comment of the author, and so should be included into the discussion section and removed from the results.

Line 104 Please change "most of them” with a precise number and percentage. 

Line 108 Better to summarize how many paper (number and percetage) analyze that method/that procedure/that technique (i.e. 3 paper out of 30; 10%; consider the use of laser: 1 on degenerative joint disease and 2 on muscular problems)

Line 110-112 and Line 114 -120. This part comprises comment from the authors. It should be included into the discussion section and not results. I suggest to revise the way in which results are proposed. 

Figure 1 Include a reference into the text. Reader is more facilitated in the compehension if the name of technique will be included into the “cake” next to the %. 

Line 122 Please include citation

Line 130 Please include citation

Line 133 The statement “the physiotherapist knows the limits of its competence” is and ambiguous sentence. This may lead me to ask if the author consider the professional figure of physiotherapist as a Doctor in Veterinary medicine or if they consider him as a technician? There is still a different consideration of the same professional figure all around Europe.

Moreover, consider to change the sentence in “….the competence of the physiotherapist must be complementary to the ones of the orthopedic/or clinician” (this is just an example).

Line 134. This is surely true. But effectiveness of physiotherapy is also related to the problem we treat. For example physiotherapy will be “curative” for some problems (i.e. rehab after  a tendinopathy or surgery) but will be less “curative” if we treat an elbow affected by severe ostheoarthrosis. So the initial diagnosis is mandatory and will lead the physiotherapist on therapeutical program.

Line 166 This should not be considered a therapy based on water. But  heat and cold (temperature) as therapeutic agent.

Line 209 I ask if it is correct to include acupuncture between the “physiotherapy techniques”. This method is based on different principles and ideas. In my opinion this is not related to the title and should not be included into this paper. 

Line 215 It is incorrect to say that acupuncture is an “effective method for treatment of cervical protrusion”. Authors should define more clearly that acupuncture can be used in the conservative management of pain related to intervertebral disc protrusions. 

Line 218 “Epilepsy and acupuncture” not related to this paper.

Line  229 In my opinion this is not “a type of physiotherapy” but a technique that the physiotherapist may use in association with other procedures/drugs/techniques. 

Line 231 This is an author’s comment? Or a citation? In this case include citation please. If we analyze the data reported into the text, we cannot state that. There is no data proposed that can lead the author to state that.

Line 236 Please include citation

Line 266 Please include citation

Line 302 Please need for more details. It is not clear what is the purpose of this statement by the authors. It is clear that the author put this point as an example. But, however, it is ambiguous to include this part in a paper concerning the review on physiatrics techniques. 

Line 312 what data may lead the author to conclude that? Which techniques the author is referring to?

Line 319 what data may lead the author to conclude that?

Line 322 Please describe what invasive methods and what restrictions.

The title of the paper is precise and punctual. 

It seems that the text is not congruous with the title.

The paper looks like a summary on the main physiotherapic techniques rather than a comparative analysis about all of them. The authors need to revise widely the results they proposed in order to make their considerations.

There is the need for more objective data.

Conclusions are not related to the discussion

I suggest to reconsider the paper for acceptance after major revision.

Author Response

Answer for the Reviewer 4

Thank you very much for your thorough review and we appreciate your valuable comments.

*Line 25 (and forward line 63). It is not clear how the term “congenital” in used. Are the authors referring to any specific disease? Need for explanations.

– Has been modified  (line 71 in the revised manuscript)

* Line 32.  In this statement it is not clear the figure of the “physiotherapist” compared to the “veterinarian".  Maybe the author meant the “clinician” instead of the veterinarian? Close cooperation is surely needed. Better to explain that the cooperation must include all the different professional figures that play a fundamental role: the orthopedic clinician, the radiologist and the physiotherapist. Moreover the fourth figure involved is the owner that have to follow the indication of the med vet physiotherapist.

- The authors believe that the term "clinician" narrows the concept of veterinarians and changes the meaning of the sentence. The vocabulary is consistent with that of the analyzed articles.

*Line 51 This is not true. The proof is The college of sports medicine and rehabilitation that is rapidly growing up and rising. It is a reality. Maybe is better to say that physiotherapy deserves more attention in consideration of its promising future. 

– Modified but this is a citation

*Line 101 This sounds like a comment of the author, and so should be included into the discussion section and removed from the results.

- In accordance with the comments of the reviewer, all content which belonged to the discussion was removed from the results.

*Line 104 Please change "most of them” with a precise number and percentage. 

 – Has been changed but it is difficult to get specific numbers because if this topic appears as a side topic in the publication, it is difficult to be classified.

*Line 108 Better to summarize how many paper (number and percetage) analyze that method/that procedure/that technique (i.e. 3 paper out of 30; 10%; consider the use of laser: 1 on degenerative joint disease and 2 on muscular problems)

- Such a juxtaposition would result in a very large chapter of results, which is not what this manuscript was about. The intention of the authors was to highlight the discussion and describe the physiotherapeutic methods, not to focus on statistics. Moreover, any statistical comparison is inadequate due to the use of different research methods within one physiotherapy technique and different sizes of research groups.

*Line 110-112 and Line 114 -120. Line 110-112 and Line 114 -120. This part comprises comment from the authors. It should be included into the discussion section and not results. I suggest to revise the way in which results are proposed. 

– Have been removed

*Figure 1 Include a reference into the text. The reader is more facilitated in the comprehension if the name of technique is included in the “cake” next to the %. 

- Such a modification would make the graph unreadable (figure 2 in the revised manuscript)

*Line 122, 130 -  Some of the sentences have not been changed because the sentences are not quotations, but the result of the authors' experience. Such wording appears in many publications and is not quoted

*Line 133 The statement “the physiotherapist knows the limits of its competence” is and ambiguous sentence. This may lead me to ask if the author consider the professional figure of physiotherapist as a Doctor in Veterinary medicine or if they consider him as a technician? There is still a different consideration of the same professional figure all around Europe. Moreover, consider to change the sentence in “….the competence of the physiotherapist must be complementary to the ones of the orthopedic/or clinician” (this is just an example).

– Has been changed

*Line 134. This is surely true. But effectiveness of physiotherapy is also related to the problem we treat. For example physiotherapy will be “curative” for some problems (i.e. rehab after  a tendinopathy or surgery) but will be less “curative” if we treat an elbow affected by severe ostheoarthrosis. So the initial diagnosis is mandatory and will lead the physiotherapist on therapeutical program.

– This is clarified in the further text.

*Line 166 This should not be considered a therapy based on water. But  heat and cold (temperature) as therapeutic agent.  

– Has been changed

*Line 209 I ask if it is correct to include acupuncture between the “physiotherapy techniques”. This method is based on different principles and ideas. In my opinion this is not related to the title and should not be included into this paper. 

- In many publications, acupuncture is referred to as a method of physiotherapy. In older articles it was indeed separated, but the article analyzes publications after 2000 where it is included as a technique of physiotherapy.

*Line 215 It is incorrect to say that acupuncture is an “effective method for treatment of cervical protrusion”. Authors should define more clearly that acupuncture can be used in the conservative management of pain related to intervertebral disc protrusions. 

– Has been modified

*Line 218 “Epilepsy and acupuncture” not related to this paper

- This article was chosen because it describes acupuncture as a technique of physiotherapy used in the treatment of human epilepsy. We write about various techniques for various diseases to emphasize their potential and the possible possibility of transferring them to animals.

*Line  229 In my opinion this is not “a type of physiotherapy” but a technique that the physiotherapist may use in association with other procedures/drugs/techniques

- This term is used in many articles, but if the reviewer wishes, we can change it to technics

*Line 231 This is an author’s comment? Or a citation? In this case include citation please. If we analyze the data reported into the text, we cannot state that. There is no data proposed that can lead the author to state that.

– Has been changed

*Line 236 – authors speaking from their experience, this is not a citation

*Line 266 – is related to article 14 (citation 14)

*Line 302 Please need for more details. It is not clear what is the purpose of this statement by the authors. It is clear that the author put this point as an example. But, however, it is ambiguous to include this part in a paper concerning the review on physiotherapy techniques.

-  This sentence introduces the next paragraph and is an example.

*Line 312 what data may lead the author to conclude that? Which techniques the author is referring to?

- This is a general introduction to the covered techniques of physiotherapy . It is difficult to describe which method was introduced in which year and adapted by whom because this data is not available.

*Line 319 what data may lead the author to conclude that? 

- We compare the data from the articles used for the manuscript with the authors' observations and the opinion of the physiotherapeutic community. We point out that there is a potential gap in research and duplicate articles that most often describe techniques that are easier to study (non-invasive testing due to the status of a companion animal), while the publication could take into account the frequency of a given method, which reflects effectiveness.

Line 322 Please describe what invasive methods and what restrictions. 

- The dog is a companion animal and it is not allowed to conduct any experiments on it (in Poland). The Ethics Committee may agree to experiments on dogs only and exclusively from laboratory breeding. There are different legal regulations in different countries, but in the vast majority of them the status of a companion animal is associated with the impossibility of conducting experimental tests, and only tests within the framework of veterinary practice (treatment and prophylaxis) are allowed.

The discussion has been modified, however, some of the sentences in the discussion result from the experience and practice of the authors and general observations. Hence, it is not always possible to provide a citation.

As suggested by the reviewers, the title has been modified to better reflect the content of the article.

Round 2

Reviewer 2 Report

Thank you for the opportunity to re-review. I can see that attempts have been made to refine the manuscript. 

Many of my concerns regarding the methods and results sections persist, and have not been adequately addressed. 

Method:

Provide details of methods used to grade the quality of included studies.

Provide details of method used to assess studies for risk of bias. That the authors are experienced experts in the field is not a method, nor sufficient rationale.

Outline method used for data extraction from included studies, as well as any planned statistical analysis or meta-analysis. 

Results

Add results of study quality grading and risk of bias assessment.

Add a table or summary of the effect sizes / outcomes identified with the use of the physiotherapy methods reported in the included studies. The current results show which techniques are commonly used, but does not indicate whether they are effective. 

Much of the content in the discussion actually belongs in the results. In a systematic review emphasis is given to the most robust conclusions, which are those that can be drawn from the aggregation of data across systematically sourced studies. Assertions drawn from single sources are afforded less weight. Please re-write this section to clarify how many studies inform each assertion, and provide details as to how many participants were included in each study, the procedures undertaken, and the outcomes obtained. Then add an overall statement or table to summarise the key findings from your review. 

Once this expanded results section is complete, you may wish to add a short discussion to place the results in the broader context of the field of study. 

Finally, please carefully proof-read your manuscript. The editing already undertaken has introduced some minor errors. 

Author Response

Answer to Reviewer 2

Thank you very much for your thorough review and we appreciate your valuable comments. The recommended changes were introduced in accordance with the reviewer's comments.

*Provide details of methods used to grade the quality of included studies - done

*Provide details of method used to assess studies for risk of bias. That the authors are experienced experts in the field is not a method, nor sufficient rationale.-  done

*Outline method used for data extraction from included studies, as well as any planned statistical analysis or meta-analysis. - No statistics has been applied, extraction of the date has been explained

*Add results of study quality grading and risk of bias assessment. - done

*Add a table or summary of the effect sizes / outcomes identified with the use of the physiotherapy methods reported in the included studies. The current results show which techniques are commonly used, but does not indicate whether they are effective. - done

*Much of the content in the discussion actually belongs in the results. In a systematic review emphasis is given to the most robust conclusions, which are those that can be drawn from the aggregation of data across systematically sourced studies. Assertions drawn from single sources are afforded less weight. Please re-write this section to clarify how many studies inform each assertion, and provide details as to how many participants were included in each study, the procedures undertaken, and the outcomes obtained. Then add an overall statement or table to summarise the key findings from your review.   - done

*Once this expanded results section is complete, you may wish to add a short discussion to place the results in the broader context of the field of study. - done

*Finally, please carefully proof-read your manuscript. The editing already undertaken has introduced some minor errors. - Re-reviewed by the translator

Reviewer 3 Report

The suitability of the main title looks better. The paper is adequate and followed many of the recommendations, but I insist that the discussion be revised and that authors be inserted throughout the text.

Author Response

Answer to Reviewer 3

Thank you very much for your thorough review and we appreciate your valuable comments. The recommended changes were introduced in accordance with the reviewer's comments.

The suitability of the main title looks better. The paper is adequate and followed many of the recommendations, but I insist that the discussion be revised and that authors be inserted throughout the text. – discussion has been modified. If there are still comments to the discussion, please specify the passages where the authors and quotations should be inserted.

Reviewer 4 Report

*Line 108 Better to summarize how many paper (number and percetage) analyze that method/that procedure/that technique (i.e. 3 paper out of 30; 10%; consider the use of laser: 1 on degenerative joint disease and 2 on muscular problems)

- Such a juxtaposition would result in a very large chapter of results, which is not what this manuscript was about. The intention of the authors was to highlight the discussion and describe the physiotherapeutic methods, not to focus on statistics. Moreover, any statistical comparison is inadequate due to the use of different research methods within one physiotherapy technique and different sizes of research groups.

***I understand, however is authors want to discuss they have to start from results data. I believe that in a scientific paper  the results must be presented in the most accurate way. Even if this means to write a long result section. Moreover if you want to write a "Systematic review" this is necessary.

*Line 209 I ask if it is correct to include acupuncture between the “physiotherapy techniques”. This method is based on different principles and ideas. In my opinion this is not related to the title and should not be included into this paper. 

- In many publications, acupuncture is referred to as a method of physiotherapy. In older articles it was indeed separated, but the article analyzes publications after 2000 where it is included as a technique of physiotherapy.

*** Acupuncture is a different technique based on different principles and beliefs. Physiotherapy technique includes mechanical or nonmechalnichal techniques that can be an instrument to reach the purpose of the physiatric therapy. Acupuncture in not a technique, is a different alternative medicine based of different beliefs. If we do acupuncture we are not doign physiotheraphy but acupuncture. 

*Line 218 “Epilepsy and acupuncture” not related to this paper

- This article was chosen because it describes acupuncture as a technique of physiotherapy used in the treatment of human epilepsy. We write about various techniques for various diseases to emphasize their potential and the possible possibility of transferring them to animals.

***This points out that this is not related to the purpose of the paper. 

*Line  229 In my opinion this is not “a type of physiotherapy” but a technique that the physiotherapist may use in association with other procedures/drugs/techniques

- This term is used in many articles, but if the reviewer wishes, we can change it to technics

***yes please

*Line 312 what data may lead the author to conclude that? Which techniques the author is referring to?

- This is a general introduction to the covered techniques of physiotherapy . It is difficult to describe which method was introduced in which year and adapted by whom because this data is not available.

***So, if author do not have enough data to prove that, it' is better to not include this conclusion into the paper.

Author Response

Answer to Reviewer 4

Thank you very much for your thorough review and we appreciate your valuable comments. The recommended changes were introduced in accordance with the reviewer's comments.

*Line 108 Better to summarize how many paper (number and percetage) analyze that method/that procedure/that technique (i.e. 3 paper out of 30; 10%; consider the use of laser: 1 on degenerative joint disease and 2 on muscular problems)

- Such a juxtaposition would result in a very large chapter of results, which is not what this manuscript was about. The intention of the authors was to highlight the discussion and describe the physiotherapeutic methods, not to focus on statistics. Moreover, any statistical comparison is inadequate due to the use of different research methods within one physiotherapy technique and different sizes of research groups.

***I understand, however is authors want to discuss they have to start from results data. I believe that in a scientific paper  the results must be presented in the most accurate way. Even if this means to write a long result section. Moreover if you want to write a "Systematic review" this is necessary. done

 *Line 209 I ask if it is correct to include acupuncture between the “physiotherapy techniques”. This method is based on different principles and ideas. In my opinion this is not related to the title and should not be included into this paper. 

- In many publications, acupuncture is referred to as a method of physiotherapy. In older articles it was indeed separated, but the article analyzes publications after 2000 where it is included as a technique of physiotherapy.

*** Acupuncture is a different technique based on different principles and beliefs. Physiotherapy technique includes mechanical or nonmechalnichal techniques that can be an instrument to reach the purpose of the physiatric therapy. Acupuncture in not a technique, is a different alternative medicine based of different beliefs. If we do acupuncture we are not doign physiotheraphy but acupuncture. 

- The authors do not classify acupuncture as a physiotherapeutic method, but are in the opinion that it is an adjunctive method which is on the border of physiotherapy and alternative medicine, and it is a therapy supporting classic therapeutic methods - it has been corrected in the text and placed in broader context as alternative medicine

*Line 218 “Epilepsy and acupuncture” not related to this paper

- This article was chosen because it describes acupuncture as a technique of physiotherapy used in the treatment of human epilepsy. We write about various techniques for various diseases to emphasize their potential and the possible possibility of transferring them to animals.

***This points out that this is not related to the purpose of the paper. 

- It is an article that supports the treatment of epilepsy in humans. In dogs, epilepsy is a big problem and is relatively common, especially in certain breeds of dogs. The intention of the authors was to draw attention to the fact that, as a technique effectively used in the treatment of humans, it has the potential to be transferred to dogs. Even though these techniques are used in dogs, there is no published research on the subject.

*Line  229 In my opinion this is not “a type of physiotherapy” but a technique that the physiotherapist may use in association with other procedures/drugs/techniques

- This term is used in many articles, but if the reviewer wishes, we can change it to technics

***yes please has been changed

*Line 312 what data may lead the author to conclude that? Which techniques the author is referring to?

- This is a general introduction to the covered techniques of physiotherapy. It is difficult to describe which method was introduced in which year and adapted by whom because this data is not available.

***So, if author do not have enough data to prove that, it' is better to not include this conclusion into the paper. - done

Round 3

Reviewer 2 Report

Thank you for your considerable revision of this manuscript.